# The Atmospheric Influence on Cosmic-Ray-Induced Ionization and Absorbed Dose Rates

Alexandre Winant [1,2,*], Viviane Pierrard [1,2], Edith Botek [1] and Konstantin Herbst [3]

1 Space Physics Department, Royal Belgian Institute for Space Aeronomy (BIRA-IASB), 1180 Brussels, Belgium; viviane.pierrard@aeronomie.be (V.P.); edith.botek@aeronomie.be (E.B.)

2 Center for Space Radiations, Earth and Life Institute ELI-C, Université Catholique de Louvain, 1348 Louvain-La-Neuve, Belgium

3 Institut für Experimentelle und Angewandte Physik, Christian-Albrechts-Universität zu Kiel, 24148 Kiel, Germany; herbst@physik.uni-kiel.de

* Correspondence: alexandre.winant@aeronomie.be

**Abstract:** When high-energy particles originating from space penetrate the atmosphere, they may interact with atoms and molecules, initiating air showers composed of secondary and tertiary particles propagating towards the ground. They can cause ionization of the atmosphere and contribute to the radiation dose at low altitudes. This work uses the GEANT-4-based Atmospheric Radiation Interaction Simulator (AtRIS) toolkit to compute these quantities in the Earth's atmosphere. We take advantage of the unique Planet Specification File (PSF) of the Atmospheric Radiation Interaction Simulator (AtRIS) to investigate the effect of the state of the atmosphere on the resulting induced ionization and absorbed dose rates from the top of the atmosphere (at 100 km) down to the surface. The atmospheric profiles (density, pressure, temperature, and composition) are computed with the NRLMSISE-00 model at various latitudes and for every month of 2014, corresponding to the last maximum of solar activity. The resulting ionization and dose rates present different profiles that vary with latitude in the atmosphere, with the relative difference between equatorial and high latitude ionization rates reaching 68% in the Pfotzer maximum. We obtain differences of up to 59% between the equator and high latitudes observed at commercial flight altitudes for the radiation dose. Both ionization and absorbed dose rates also feature anti-phased seasonal variations in the two hemispheres throughout 2014. Based on these results, we computed global maps of the ionization and dose rates at fixed altitudes in the atmosphere by using precomputed maps of the effective vertical cutoff rigidities and the results of three AtRIS simulations to consider the effect of latitude. While sharing the same general structure as maps created with a single profile, these new maps also show a clear asymmetry in the ionization and absorbed dose rates in the polar regions.

**Keywords:** ionization; absorbed dose; galactic cosmic rays; atmosphere

## 1. Introduction

When high-energy charged cosmic ray (CR) particles, forming the radiative environment of the Earth, reach the atmosphere, they cause ionization at lower altitudes through complex nucleon–muon–electromagnetic cascades [1–3] contributing to the radiation dose impacting human health [4]. The main contributors to CR-induced ionization (CRII) and radiation dose in the atmosphere are the galactic cosmic rays (GCRs) originating from far outside the solar system. GCRs are mainly composed of protons and helium nuclei accelerated to energies from about $10^7$ eV up to $10^{21}$ eV [5] constantly and omnidirectionally bombarding the Earth. As they propagate through the heliosphere, GCRs interact with the plasma of the solar wind expanding outward from the Sun and the Interplanetary Magnetic Field (IMF) [6]. The intensity of GCRs is modulated by the 11-year solar cycle by means of an anti-correlated dependence [7]. GCRs are a background source of radiation that constantly ionizes the Earth's atmosphere. However, transient radiation variations

can occur during strong solar events, such as flares and Coronal Mass Ejections (CMEs) [8]. During such events, solar protons are accelerated to energies from hundreds of keV to tens of GeV, which then form an important transient source of particle radiation, the solar energetic particles (SEPs), which can impact the Earth's atmosphere [9]. SEPs have a different composition and lower energies than GCRs, and thus, their contribution to atmospheric ionization and radiation dose takes place at higher altitudes [3]. However, in the case of so-called Ground Level Enhancement (GLE) events, strong solar events that can reach the ground, protons can increase atmospheric ionization down to 20 km [10].

The induced ionization by space radiation has been and still is heavily investigated because of its impact on the physical and chemical processes in the atmosphere. Indeed, it has been found that CRII can influence the global atmospheric electricity, with enhanced vertical current density during strong SEP events [11,12]. CRII also plays a crucial role in atmospheric chemistry, especially in ozone depletion [13], cloud formation, cloud coverage, and precipitation [3,14–17]. Many models have been developed to compute the ionization caused by cosmic rays in the atmosphere, such as PLANETOCOSMICS [18], CRAC:CRII [19], and others [20,21]. Recently, computations of the ionization rate were performed for the entire surface of the Earth at various altitudes during quiet periods and also for SEP events [22,23]. In addition to contributing to atmospheric ionization, cosmic rays lead to radiation exposure in the atmosphere. These radiation fields have been recognized as a hazard to the health of aircraft crews and frequent fliers, especially during SEP events [24]. For this reason, many efforts have also been directed toward calculating the radiation dose caused by GCRs in the atmosphere [25–27].

In this work, we compute the CR-induced ionization and the absorbed dose rate in the atmosphere with the Atmosphere Radiation Interaction Simulator (AtRIS) [28,29]. We compute and compare these quantities throughout the entire atmosphere (between 0 km and 100 km) with various atmospheric input profiles for the AtRIS model to assess the importance of the state of the atmosphere on the resulting ionization and dose rates. We investigate the latitudinal dependence and the impact of seasonal variations on the dose rates and the ionization. The input atmospheres are either taken at a fixed date at various latitudes or at fixed latitudes for different months of 2014, corresponding to the maximum solar activity. In addition, global maps of the ionization and dose rates are presented for January and August 2014. These maps are computed by using seven atmospheric input profiles (covering high, medium, and low latitudes). The second section of this paper describes the model we use and the simulation setup. Moreover, we describe the procedure we followed to compute the maps presented below. Section 3 presents and discusses the results of our simulations, starting with the effect of the latitude on the ionization and dose, followed by their seasonal variability. Then, global maps of the ionization and absorbed dose are presented. Finally, in Section 4, we conclude the importance of the input atmosphere for the computation of CRII and radiation dose in Earth's atmosphere.

## 2. Materials and Methods

### 2.1. Model

We use AtRIS to compute the ionization [28] and absorbed [29] dose rates induced in the atmosphere by GCRs, which has been validated for Earth [28], Mars [11,30], and Venus [31,32]. As with other models that serve the same purpose (e.g., [18,20,33]), AtRIS is based on Monte Carlo simulations and, more precisely, on the GEANT-4 toolkit [34] developed by the Central European Research Network (CERN). In this study, we took advantage of the Planet Specification File (PSF) featured in AtRIS, which allows the user to build any planet and its atmosphere element by element. We used this specific feature to simulate Earth's atmosphere, varying the density, pressure, temperature, and composition profiles over different latitudes and dates to investigate how the state of the atmosphere affects the ionization and the dose rates induced by cosmic rays. In our simulation setup, the atmosphere is 100 km thick and is composed of 200 layers, 500 m thick each. In addition, the atmosphere is split into three *sub-atmospheres*, three latitude bands, from [90° N to 30° N],

[30° N to 30° S], and [30°S to 90°S]. In doing so, we defined the geometry of the atmosphere in which AtRIS propagates energetic particles. However, atmospheric density, temperature, pressure, and composition profiles for each *sub-atmosphere* need to be specified. To this end, the atmosphere model NRLMSISE-00 [35] (accessible at https://ccmc.gsfc.nasa.gov/modelweb/models/nrlmsise00.php, accessed on 27 November 2023) is used. Thus, for each *sub-atmosphere*, NRLMSISE-00 provides the input atmospheric profiles for the simulation (see Figure 1). In the present work, we only change two parameters of the NRLMSISE-00 model, the latitudes and the date at which the model is run. The first objective with this setup was to create a more realistic model of the atmosphere for the simulation. However, the main benefit of this approach is that it allows us to simultaneously compute the ionization and dose rate profiles for three different input atmospheres. This makes the overall process faster than running three simulations with a single input atmosphere.

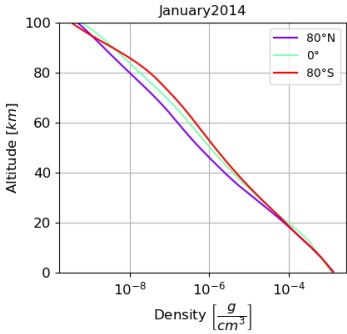

**Figure 1.** Atmospheric density profiles computed by NRLMSISE-00 in January 2014, at 80° N, 0°, and 80° S. These profiles are used to compute the atmospheric response matrices displayed in Figure 2.

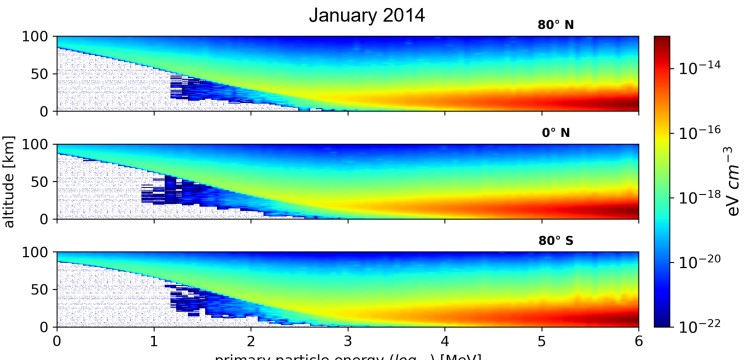

**Figure 2.** Unscaled ionization atmospheric response matrix (ARM) for three different atmospheric profiles at three latitudes in January 2014. Impacting primary particles are protons, with energies ranging from 1 MeV to 1 TeV in logarithmically equidistant energy bins.

AtRIS calculates the average impact of a single primary particle with a given energy at different altitudes [28,29] by considering the effect of the primary particle, as well as the effect of all the resulting secondary particles (see [28] for further detail). The resulting matrices describe the relation between the primary particle energy and the altitude-dependent average ionization in the atmosphere. Thus, with this set up, for which the atmosphere is divided into three *sub-atmospheres*, three atmospheric response matrices (ARMs) were computed in a single simulation. Such atmospheric response matrices computed at 80° N , 0°, and 80° S are shown in Figure 2 from top to bottom, respectively. Note that such matrices only describe how much a particle (in this case, a proton) of a given energy (between 1 MeV and 1 TeV) contributes on average to the ionization of the atmosphere if it is able to reach the top of the atmosphere (TOA). At this point, the flux of primary particles as a function of their energy is not taken into account, nor is the effect of the geomagnetic field, so that

the differences between these matrices is only due to changes in the atmospheric input of the simulations.

For the radiation dose calculations, AtRIS performs an on-the-fly computation for specific phantoms whose response to ionizing particles is known and stored in look-up tables [29,31,36]. For a given phantom, the corresponding look-up table contains the relative ionization efficiency for each particle that contributes to the dose in the phantom in an energy range from 1 eV to 10 TeV. The relative ionization efficiency is defined as the ratio between the energy deposited in the phantom by a particle and the kinetic energy of the particle. When a simulation with AtRIS is performed, the absorbed dose in the phantom is thus computed by multiplying the ionizing particle energy with its corresponding relative ionization efficiency. At a given altitude and for a given energy of the primary particle energy, the contributions to the dose of each particle in the atmospheric cascade are added together. Repeating this process for each atmospheric layer and each energy bin of the primary particles, an ARM for the absorbed dose is constructed [29]. One phantom provided in AtRIS is an ICRU sphere (defined by the International Commission on Radiation Units). However, since AtRIS also aims to be applicable to exoplanetary studies and the purpose of the radiation computations is to assess exoplanetary habitability, the ICRU sphere implemented in AtRIS is made out of water rather than Tissue Equivalent Material. All the available look-up tables for existing phantoms can be found through the following link: https://zenodo.org/records/3633451, (accessed on 27 November 2023).

The flux of primary GCRs at the TOA is computed using the solution of the force-field approximation [6,37,38] to solve the transport equation of charged particles through the heliosphere introduced by Parker [39]:

$$\frac{dJ}{dE}(E, \Phi) = J_{LIS}(E + \Phi)\frac{E(E + \Phi)}{(E + \Phi)(E + \Phi + 2E_r)},$$　　　　(1)

where $J_{LIS}(E)$ is the differential Local Interstellar Spectrum (LIS) of primary particles before they interact with the heliosphere, $E_r$ is the rest energy, and $\Phi = (Ze/A)\phi(t)$ is the solar modulation function with $Z$ the electric charge, $e$ the charge of the electron, $A$ the atomic number, and $\phi$ is modulation potential. The solution of the equation allows the differential flux of GCRs at 1 AU to be described with only one free parameter, the modulation potential $\phi(t)$ in units of an electric potential (eV). Thus, the differential flux of primaries at Earth only depends on the particle type, the solar activity, and the LIS, which is parameterized based on numerical simulations and fitting observations for protons [40–42] and for electrons [43,44]. In the following, we use the LIS proposed by Herbst et al. [42] for protons and alphas, taking Voyager I data into account as follows:

$$J_{LIS}(E) = \begin{cases} 0.707 \exp(4.64 - 0.036(\ln E)^2 - 2.91\sqrt{E}) & \text{if } E < 1.4\,\text{GeV} \\ 0.685 \exp(3.22 - 2.78(\ln E) - 1.5/E) & \text{if } E \geq 1.4\,\text{GeV} \end{cases}$$　　　(2)

In this initial study, we only consider primary particles of cosmic origin, which feature a wide range of energies and are responsible for the ionization at low altitudes. More specifically, primary particles in our simulations are GCR protons and alphas with energies ranging between 1 MeV and 1 TeV, divided in 100 bins. In addition, in order to take into account the effect of the Earth's geomagnetic field, we compare the rigidity of the primary particle ($R = pc/q$) and the effective vertical rigidity cutoff ($R_c$) at a given location on the Earth. If for a given particle $R < R_c$, then the particle will not be able to reach the TOA. Thus, the flux of the primary particle contributing to the ionization and the dose is effectively given by: $\frac{dJ}{dE}(E(R), \Phi)$ for $R \geq R_c$. The conversion between particle energy and rigidity is computed as follows,

$$R = \frac{A}{Z}\sqrt{E^2 + 2EE_0},$$　　　　(3)

where *A* is the mass number, *Z* is the charge, $E_0$ is the rest energy per nucleon, and *E* is the particle's kinetic energy per nucleon. With our present setup, with 100 energy bins ranging from 1 MeV to 1 TeV and with an atmospheric layer thickness of 0.5 km, the maximum estimated statistical error on the ionization and absorbed dose rate profiles is 13%.

### 2.2. Global Maps' Computation

We computed new global maps of the ionization rate and absorbed dose rate induced in the atmosphere by GCRs at fixed altitudes, covering the entire surface of the Earth. In order to provide more accurate maps of the ionization and absorbed dose rates, only considering high latitude profiles and the equator is not enough. Ionization and dose rates at mid and low latitudes are thus also computed. Due to computation limitations, it was not possible to run a single simulation in which the atmosphere was divided in seven *sub-atmospheres* representing high (80°), mid (45°), and low (15°) latitudes and the equator. In order to compute the response of the atmosphere to GCRs at these latitudes in both hemispheres, the simulations with three *sub-atmospheres* were repeated three times, each time changing the latitude of the input atmosphere.Note that for the maps presented in the following section, the ionization and dose rates at latitudes higher than 80° are computed with the results of the simulations whose input atmosphere is taken at 80°.

Thus, atmospheric ionization rate and dose rate profiles are evaluated at six latitudes covering high, mid, and low latitudes. Note that each simulation contains information on the ionization and absorbed dose in the equatorial atmosphere. In total, by taking the results at the equator, the ionization and dose rates are computed for seven input atmospheric profiles. Moreover, to take into account the effect of the geomagnetic field on the primary particle spectra, we use global maps of the effective vertical cutoff rigidities computed with the cutoff rigidity (COR) model [45] (accessible at https://cor.crmodels.org/geo-mag/, accessed on 27 November 2023), which uses the International Geomagnetic Reference Field (IGRF) as input.

To make the computation of the maps faster, some preprocessing is conducted with the results of AtRIS. Because the cutoff rigidity values at the surface of the Earth change at every point at the surface, the ionization and dose rates must be computed for cutoff rigidities between 0 GV and 18 GV. Since the resolution of the COR model is set to 0.01 GV, the ionization and dose rate profiles are computed with the same resolution for each input latitude. In addition, primary particles are affected by the solar modulation, so we also compute the dose and ionization rate profiles for different values of the modulation potential (in this case, from 0 MV to 1500 MV with a 100 MV resolution). We applied this procedure for each simulated latitude and stored the resulting data in separate files. To make a global map of the ionization rate or absorbed dose rate, a value of $\phi$ must be fixed together with an altitude. Thus, when these values are fixed, the ionization/dose rate variation as a function of the cutoff rigidity is known for each simulated latitude. Since computing a global map of ionization and dose rate with only seven latitudes would result in large discontinuities when transitioning from one simulation result to another, for each value of the cutoff rigidity, the ionization/dose rates are interpolated on the latitudes with a 1° resolution.

The last step to obtain the maps is to assign the ionization/dose rate, corresponding to the cutoff rigidity at the surface of the Earth for each longitude, at each degree in latitude (from 90° N to 90° S).

### 3. Results

As described above, multiple simulation runs were performed. A comparison of the results shows that the state of the atmosphere has a non-negligible impact on the resulting induced ionization and absorbed dose rate profiles.

*3.1. Ionization and Dose Rates for Atmospheres Taken at Different Latitudes*

To investigate the effect of the latitude at which the input atmosphere is taken on the resulting ionization and dose rate profiles, we performed simulations at the equator, $\pm 15°$, $\pm 45°$, and $\pm 80°$, so that low, mid, and high latitudes were taken into account. Figure 3 shows the computed ionization and absorbed dose rate profiles during the northern hemispheric winter of January 2014, as well as the atmospheric density profiles, which served as input for those computations. It is important to note that, since we only want to capture the effect of the input atmospheric profiles on the resulting ionization and dose profiles, the value of the modulation potential and the vertical cutoff rigidity are set to 700 MV and 0 GV, respectively, and are maintained constant in this section's figures. The modulation potential value was chosen to represent the averaged value over 2014. According to the cosmic ray database, the results of which are based on work in [46], the averaged modulation potential over this year was 715 MV, which was rounded down to 700 MV because the maps presented further on in this research were only computed with a 100 MV resolution. Fixing the cutoff rigidity value to 0 GV, implies that the geomagnetic field influence on the TOA primary particle's fluxes is effectively removed from the simulations. Thus, in the following, latitudinal changes in the ionization and dose rates are the results of the latitudinal changes in the atmospheric profiles that serve as an input for the simulations. Because the effect of the geomagnetic field cannot be ignored in order to model realistic values of the ionization and dose rates, the global maps of these quantities presented later in the paper are based on the values of the cutoff rigidity predicted by the COR model.

The top panels of Figure 3 display the density profiles, as predicted by the NRLMSISE-00 model, of the latitude-dependent atmospheric profiles in January 2014 (panel a) and the relative difference between the equatorial and various density profiles (panel b). Although the atmospheric density profiles display an exponential decrease with increasing altitude for all latitudes, clear density differences can be observed. The main density differences in the atmosphere occur at altitudes between 30 km and 90 km, and are at their largest between the northern and southern polar latitudes. At these altitudes, the density steadily decreases from the high southern latitudes to the high northern latitudes. At altitudes below 30 km, the relative difference between atmospheric density profiles is not the same as at higher altitudes (see panel b). Between 30 km and 20 km, atmospheric density is at its largest at low altitudes and at the equator, while it is at its minimum value in the polar regions. It is more evident when looking at the relative difference profiles. Below 10 km, we can observe an opposite behaviour, where the maximum value of density takes place in the polar atmospheres.

For all atmospheres evaluated at different latitudes, the ionization profiles share similar general features. The CR-induced ionization increases as altitude decreases from the top of the atmosphere down to the Pfotzer maximum (i.e., the region where the ionization rate is at its maximum in the atmosphere) at $\sim$15 km (depending on location) and then decreases down to the surface. Nevertheless, some differences arise as latitude changes, similar to the atmospheric density profiles: Between 90 km and 25 km, in the southern hemisphere at high latitudes (i.e., 80° S), ionization is at its maximum value and gradually decreases as latitude increases to high latitudes in the northern hemisphere. At lower altitudes and higher altitudes in the thermosphere, variations in the ionization rates with latitude are not as straightforward. Below 13 km, the ionization rate is at its minimum at the equator down to the surface and starts to increase as latitude increases in both hemispheres. However, the rise in the ionization rates does not occur at the same latitudes when comparing the northern and southern hemispheres. In the northern hemisphere, the ionization rate increases faster and reaches a maximum value at 80° N. In the Pfotzer maximum, the largest ionization rates are reached at the equator and at low latitudes. In this case, ionization variations with respect to latitude show an asymmetric behavior between the two hemispheres. In January 2014, in the Pfotzer maximum at high latitudes, the highest ionization levels were reached in the northern hemisphere. From Figure 3c, it also becomes

clear that as latitude changes, so does the altitude of the Pfotzer maximum, which is at its highest at low latitudes and the equator and its lowest at high southern latitudes.

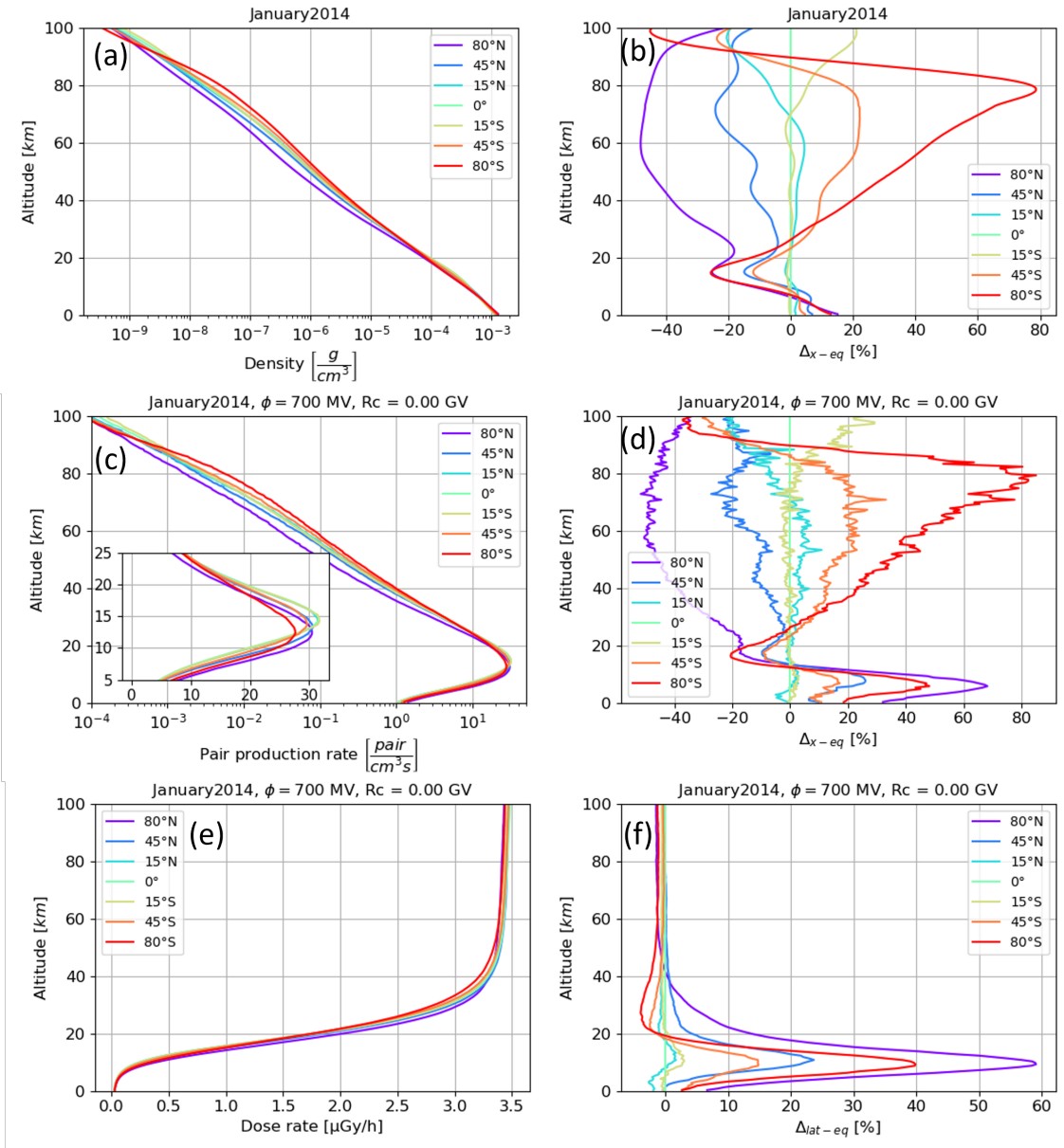

**Figure 3.** This figure shows the latitudinal variations in the atmospheric density, ionization, and absorbed dose rate profiles from 100 km of altitude down to the surface in January 2014 in panels (**a**,**c**,**e**), respectively. Atmospheric density is given in $\frac{g}{cm^3}$, the ionization rate in $\frac{pairs}{cm^3 s}$, and the absorbed dose in $\frac{\mu Gy}{h}$. In panel (**c**), the smaller graph is in linear scale and focuses on the Pfotzer maximum. Panels on the right display the relative difference in % between equatorial profiles and those computed at various latitudes. Panels (**b**,**d**,**f**) display the relative difference (in %) between the equatorial profile and profiles at other latitudes for density, ionization rate and absorbed dose rate respectively.

Regarding the relative differences between the profiles shown in panel (d) of Figure 3, the same observations can be made. This figure clearly shows that large variations between ionization profiles arise when latitude varies, especially at high altitude in the mesosphere and at altitudes below 20 km.

The largest difference between the ionization rates at the equator and high polar latitudes occurs at higher altitudes, reaching differences of 78% at around 80 km for the

southern polar atmosphere. However, if the high latitude ionization rate profiles are compared to one another (i.e., the 80° N and 80° S profiles), the relative difference increases up to 220% at 80 km (not shown in the figure). Figure 3c,d show that equatorial ionization rates at an altitude of 6 km are smaller than the ionization rates in both polar atmospheres. In January 2014, the largest relative difference is observed at 80° N, reaching a value of up to 68%. At these altitudes, the relative difference between the two high latitude profiles is reduced to 13%.

Direct correlations are visible: Between 100 km and 20 km, the seasonal and latitudinal variations in the ionization rates are very similar to those of the atmospheric density. In fact, for each altitude, a linear regression can be performed on the evolution of the density and ionization with latitude or time, and the Pearson correlation coefficient for these quantities can be computed. Between 100 km and 20 km, the correlation coefficient between the ionization rates and the atmospheric density is close to 1 (not shown here). Thus, within this altitude regime, the variations in the ionization rate with latitude and time can be directly linked to the atmospheric density variations. As energetic particles propagate through the atmosphere, they lose their energy by ionizing the ambient air, or by colliding with the molecules and atoms of the atmosphere, initiating complex secondary particle cascades (which, in turn, further ionize the atmosphere as they travel down to the ground). The mean energy loss of a heavily charged particle propagating through matter by ionization is described by the Bethe–Bloch formula. This formula, although showing great complexity, states that the energy loss of heavily charged particles is proportional to the density of electrons in the material it is propagating through [47], and thus is proportional to the density of the material itself. It increases with the density of the medium, and so does the ionization rate. Consequently, if the density is at its lowest in the upper atmosphere, as in January at 80° N, primary particles lose less energy than they do at high latitudes in the southern hemisphere. This can explain the observed seasonal and latitudinal ionization rate variations in the upper atmosphere. However, below 20 km, and particularly around the Pfotzer maximum, ionization variations differ substantially from those of the density. Still, the differences between the ionization rate profiles with latitude can be explained by the density profiles. As shown in Figure 3a, the atmospheric density differences between 10 km and 20 km are larger for the smaller latitudes. Thus, energy deposition at lower latitudes is stronger within these altitudes than in the polar regions. This causes a shift in the Pfotzer maximum to higher altitudes at regions of lower latitudes.

However, Figure 3b shows that, at these low altitudes, the density profiles at 80° N and 80° S that serve as input for the simulations are very similar, as can be seen by their same relative difference with the equatorial density, while the ionization rates differ quite significantly. This difference may be due to the fact that in the southern hemisphere, the density of the upper atmosphere is larger than in the northern hemisphere, implying that in the southern polar atmosphere (in January), ionizing particles deposit more energy in the upper atmosphere (hence, the increased ionization rate in the southern hemisphere in the upper atmosphere). Thus, the energy available to ionize the atmosphere at lower altitudes is lower than in the northern hemisphere, where less energy is lost at higher altitudes. With the same reasoning, the seasonal variations in the ionization rate can be attributed to the seasonal variations in the atmospheric density.

Figure 3e shows the absorbed dose rates. In this case, profiles at all latitudes feature similar variations with respect to the altitude, with a maximum radiation dose at high altitudes, decreasing to almost zero at the surface. However, at each latitude, these profiles differ more or less from each other at altitudes below 50 km. The left panel of Figure 3e also shows that the dose rates are at their highest (and afterwards are constant) from around 60 km up to the TOA.

Panel (f) of Figure 3 displays the altitude-dependent relative difference between the equatorial absorbed dose rates and those at various latitudes. At altitudes above 50 km, the dose rate is latitude-independent (i.e., the relative difference is lower than 1%). The largest difference relative to the equatorial absorbed dose rates occurs at ~10 km,



and reaches maximum values in the northern polar atmosphere. Although at these altitudes differences of almost 60% occur between equatorial and polar regions, this is the altitude used for commercial flights.

Variations in the radiation dose rate profiles are similar to those of the ionization rates in and below the Pfotzer maximum. At high altitudes, from 100 km to 50 km, there are no clear latitudinal differences between the dose profiles (see Figure 3f). In this case, the difference in the energy lost by ionizing particles caused by differences in the density of the upper atmosphere is sufficiently small so that the radiation dose in the water ICRU sphere is similar for all profiles. However, as the altitude decreases, the impact of the amount of energy that is lost in the upper atmosphere is increasingly important. Thus, between 20 km and 40 km, the dose rate reaches maximum values at 80° N in January and minimum ones at 80° S, since energy loss is at its largest in the southern hemisphere and smallest in the northern hemisphere. This remains true down to the surface. When considering the change in the dose rate profiles with latitude below 20 km, the dose rate becomes its minimum value at lower latitudes, while it is at its highest in the polar atmosphere. This can be explained by the Pfotzer maximum being located at higher altitudes in the tropical and equatorial atmosphere than at higher latitudes. This means that the maximum energy deposition occurs at higher altitudes at equatorial regions than in the polar atmosphere. Thus, at lower latitudes and altitudes below 20 km, the ionizing particles have lost much more energy than they would have at higher latitudes. In turn, the radiation dose is lower at equatorial regions.

*3.2. Seasonal Variations*

Further simulations were performed for equatorial and high latitude regions of both hemispheres (±80°) for every month of 2014. Figure 4a,b show the atmospheric density profiles computed with the NRLMSISE-00 model for each month of 2014 at high northern and southern latitudes, respectively. In this figure again, the exponential decrease in the density of the atmosphere with altitude remains at all times and in both hemispheres. The largest seasonal variations in atmospheric density occur at high altitudes in the atmosphere, between 90 km and 30 km in the northern hemisphere and from 90 km down to 20 km in the southern hemisphere. In this part of the atmosphere, the anti-phased variations in the density profiles at 80° N and 80° S where the maximum/minimum values of density are reached during the local summer/winter are stronger than below 20 km, where such variations are almost nonexistent.

As previously assumed, the modulation potential and the vertical cutoff rigidity values are set to 700 MV and 0 GV, respectively, in order to account only for atmospheric-induced changes in ionization and dose rates.

The left panels of Figure 5 show the ionization rate profiles as a function of time. Note that the upper altitude is restricted to 50 km, in order to focus on the Pfotzer maximum, where the ionization rate induced by GCRs is most important. A strong temporal variation in the ionization rates is visible in the high latitudes of both hemispheres (top and bottom panels). In contrast, at the equator (middle panel), the ionization rates, including the altitude of the Pfotzer maximum, which is higher up in the atmosphere than at polar regions, remain temporarily stable throughout the entire atmosphere. However, the high-latitude ionization rates of the two hemispheres show strong temporal variations at altitudes below 20 km. While in the northern hemisphere the rates are highest in February, steadily decreasing and reaching a minimum around August after which the rates increase again, the southern hemisphere ionization rates are anti-correlated. Here, the ionization rate is at its lowest in February and its highest in August. Seasonal dependencies can also be found above 20 km. These variations, however, are anti-correlated with the seasonal variations observed in the Pfotzer maximum. Thus, in a given hemisphere, as the ionization rate increases/decreases at low altitudes, it decreases/increases in the upper layers of the atmosphere.

Panels (a) and (b) of Figure 6 quantify the seasonal variations in the altitude-dependent ionization rate by showing the relative difference between the values computed for January 2014 and those computed for every month of 2014. Panel (a) displays the results at 80° N and panel (b) those at 80° S. Most of the results are similar to Figure 5, namely, the opposite seasonal variations at a given altitude between the two hemispheres and the opposite variations below and above 20 km at a given latitude. However, panels (a) and (b) further show that the intensity of these variations is not symmetric for high northern and southern latitudes. At altitudes below 20 km, the largest difference between the two ionization rate profiles occurs between January and August in both hemispheres. Nevertheless, the altitude at which the largest difference between these profiles is located is not the same at 80° N and 80° S. In the northern hemisphere, the largest variations occur below 10 km, while in the southern hemisphere, they occur at the altitudes around the Pfotzer maximum. Hence, larger maximum and lower minimum ionization rate values are observed at 80° S than at 80° N within the Pfotzer maximum (see Figure 5). At 20 km, in both the northern and the southern hemispheres, the ionization rate features almost no variations in time. The relative difference between all profiles and January is close to 0%.

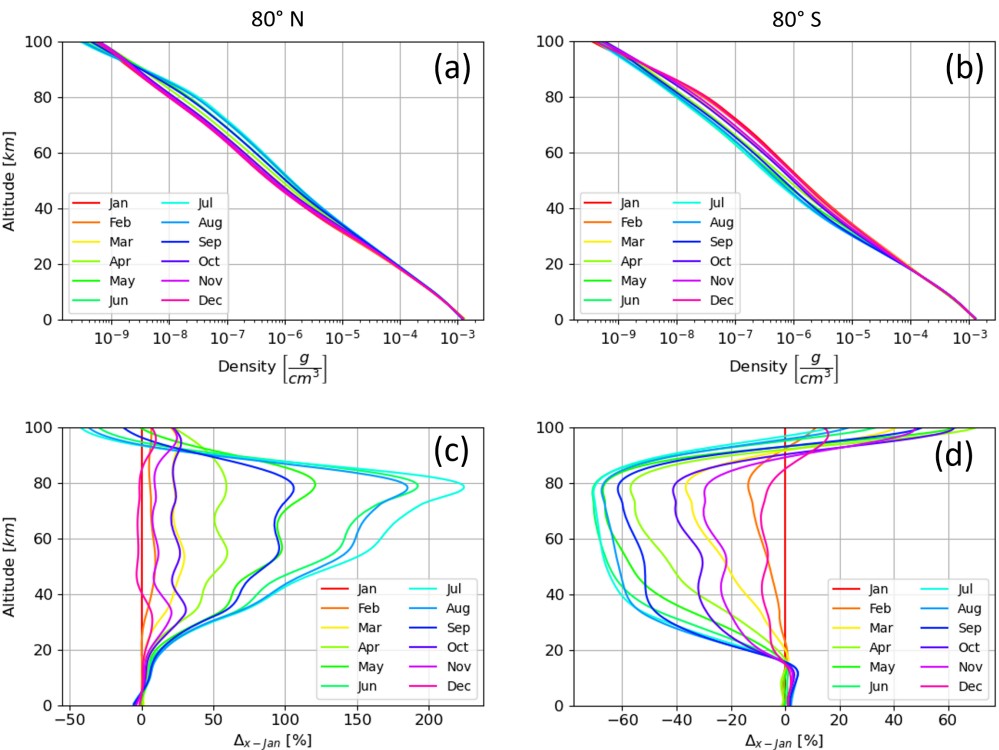

**Figure 4.** Seasonal variations of atmospheric density profiles at 80° N and 80° S as predicted by the NRLMSISE-00 model each month of 2014. Panels (**a**,**c**) respectively show the density profiles in $\frac{g}{cm^3}$ and the relative difference between January and other month density profiles at 80° N. Same for panels (**b**,**d**) but at 80° S.

The seasonal variations in the ionization rate profiles can, in this case, also be explained based on the seasonal variations in the atmospheric density profiles, in a similar way as for the latitudinal variations. From 100 km down to 20 km, the variations in the ionization rate and the variations in the atmospheric density are strongly correlated, as the loss of energy of the propagating particle is proportional to the density of the medium it is propagating through. Thus, the ionization rate at these altitudes is at its maximum/minimum during the local summer/winter as is the density of the atmosphere. Below 20 km, where density variations are very small (see Figure 4), seasonal variations in the ionization rate anti-phase with the variations observed above 20 km are explained by the greater energy loss of ionizing particles at high altitudes during the local summer in both hemispheres, leading

to a smaller amount of energy to ionize the atmosphere below 20 km and thus a minimum of ionization in the Pfotzer maximum.

A similar analysis can be made for the CR-induced absorbed dose rates. The left panel of Figure 5b shows the monthly dose rates computed throughout 2014 for a water ICRU sphere without shielding (link to AtRIS files: https://zenodo.org/records/3633451, accessed on 27 November 2023), where the altitude ranges from 0 km to 50 km. Similarly to the ionization rate, at fixed altitudes, the radiation dose rate in the atmosphere exhibits a seasonal variability that is anti-correlated at northern and southern polar latitudes, while no such seasonal variations occur at the equator. In this case, the dose rate is at its largest in January/August and its smallest in August/January at 80° N/80° S below 25 km.

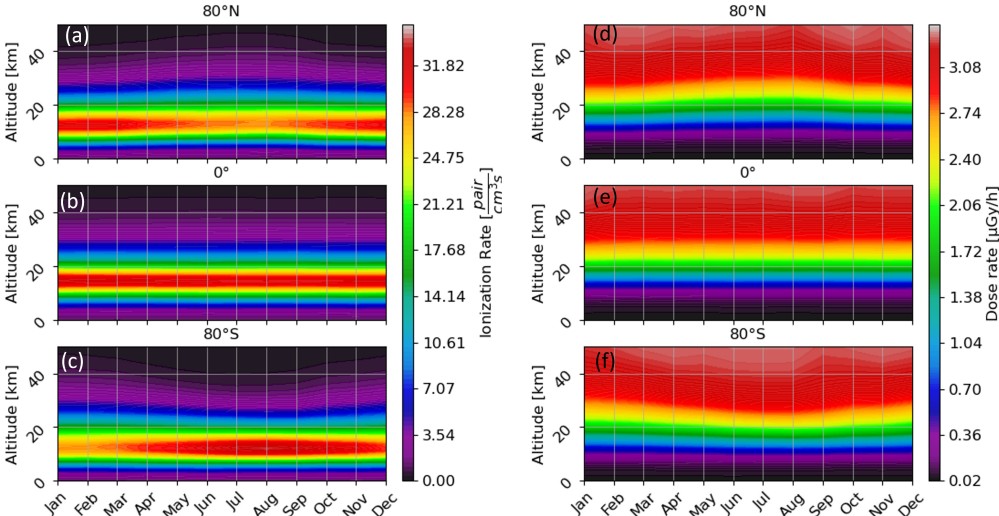

**Figure 5.** Seasonal variations in the GCR-induced ionization rate (left) and absorbed dose rate (right) in the atmosphere for 80° north for panels (**a,d**) and south for panels (**c,f**) as well as the equator for panels (**b,e**).

Panels c and d of Figure 6 are the same as panels a and b but show the relative differences between the absorbed dose rate profiles computed in January and every other month of 2014. These two panels confirm that the temporal variability in the absorbed dose in the northern hemisphere is anti-correlated with the dose variability in the southern hemisphere below 50 km. Above this altitude, there is no clear trend in the temporal evolution of the dose rates, and the different profiles remain constant up to the TOA. In addition, at such high altitudes, the relative difference between all profiles remains lower than 2.5%. Below 40 km, however, the seasonal variability is more intense in the polar atmosphere at 80° S (up to 27%), whereas it reaches −23% at 80° N (for the maximum relative difference between January and August). In the same way that the largest variations between ionization profiles occur at different altitudes below 20 km in both hemispheres, the largest variations between the dose rate profiles are also located at different altitudes at 80° N and 80° S. As for the ionization rate, this altitude is higher at southern high latitudes (18 km) and lower in the northern polar atmosphere (10 km). In this case again, seasonal variations in the absorbed dose are similar to those observed in the Pfotzer maximum for the ionization rate, and can also be explained by an increased loss of ionizing particles at high altitudes in both hemispheres due to a larger atmospheric density during the local summer. This leads to a decreased amount of energy available at lower altitudes and thus a minimum of the absorbed dose during the local summer below 40 km.

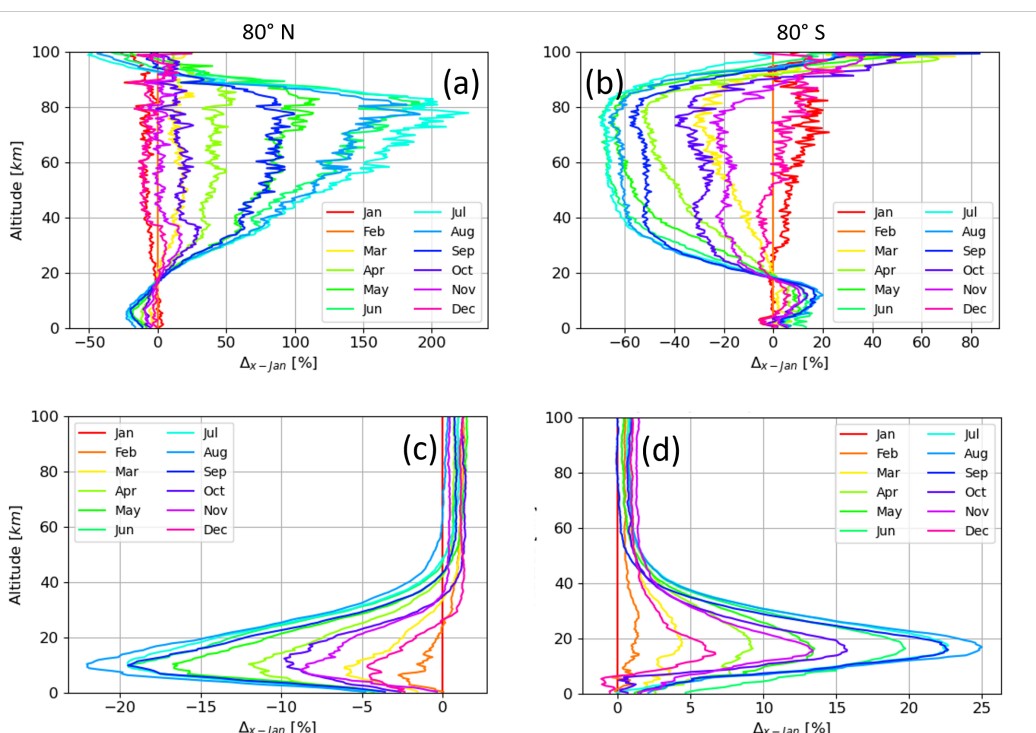

**Figure 6.** Relative difference between ionization rate profiles (**a**,**b**) and dose rate profiles (**c**,**d**) computed for January 2014 and profiles computed for each month of 2014 at 80° N (**a**,**c**) and 80° S (**b**,**d**).

### 3.3. Global Maps

Based on the results of the previous sections, it appears that the latitude and the time at which the input atmospheric model is taken may have a non-negligible influence on the resulting ionization rate and absorbed dose rate profiles at various altitudes. This remains true, to a lesser extent, even when the solar modulation and the cutoff rigidity are high. However, up until now, the effect of the geomagnetic field on the primary particle spectrum, and thus on the induced profiles, has not been considered.

Therefore, global maps of the ionization rate or dose rate were computed (following the steps described in the method section) for January and August 2014. These maps were computed with precomputed cutoff rigidity maps of the COR model. For this, the COR model needs an input geomagnetic field. This study used the International Geomagnetic Reference Field (IGRF) [48] and the outer geomagnetic field Tsyganenko (T05) [49] as input models. For the solar modulation, results from the cosmic ray database were used. In both January and August 2014, the modulation potential for the LIS model by [42] is close to 700 MV.

Figure 7 shows the resulting ionization maps at altitudes of 9 km, 12 km, and 18 km in January 2014 (left panels) and August 2014 (right panels). The computation of the ionization rate is based on the atmospheric density profiles that were presented in previous figures. The overall structure is the same in all panels: there are always higher ionization rates at high latitudes than at lower latitudes and at the equator, and an ionization minimum is present over India. These structures are caused by the Earth's geomagnetic field, which provides a better "shielding" from space radiations, where the geomagnetic field is parallel to the surface, than at high latitudes where the intensity of the geomagnetic field is higher but perpendicular to the surface, which funnels lower energy particles into the atmosphere. Thus, at low latitudes, the cutoff rigidity values are high, which means that lower energy primaries cannot reach the atmosphere, and thus, the induced ionization rate is low, while at high latitudes, all primary particles can reach the top of the atmosphere.

However, our study reveals strong hemispherical asymmetries in the polar regions at altitudes below 18 km caused by the differences in the time-dependent atmospheric profiles.

These differences are explained by the latitudinal and temporal changes in atmospheric density that were discussed in the previous sections. In January 2014, at 12 km and 9 km (panels b and c), the highest ionization rates can be observed in the polar regions of the northern hemisphere, while in August 2014 (panels e and f), it is the other way around (see also Figure 5). At altitudes above 18 km, the hemispheric asymmetry persists. However, the highest ionization rates are observed in the opposite hemisphere than the one featuring maximum rates at low altitudes (see Figure 5). In August of 2014 (panel d), however, a localized maximum is present, reaching down to latitudes of 50° in the northwestern hemisphere.

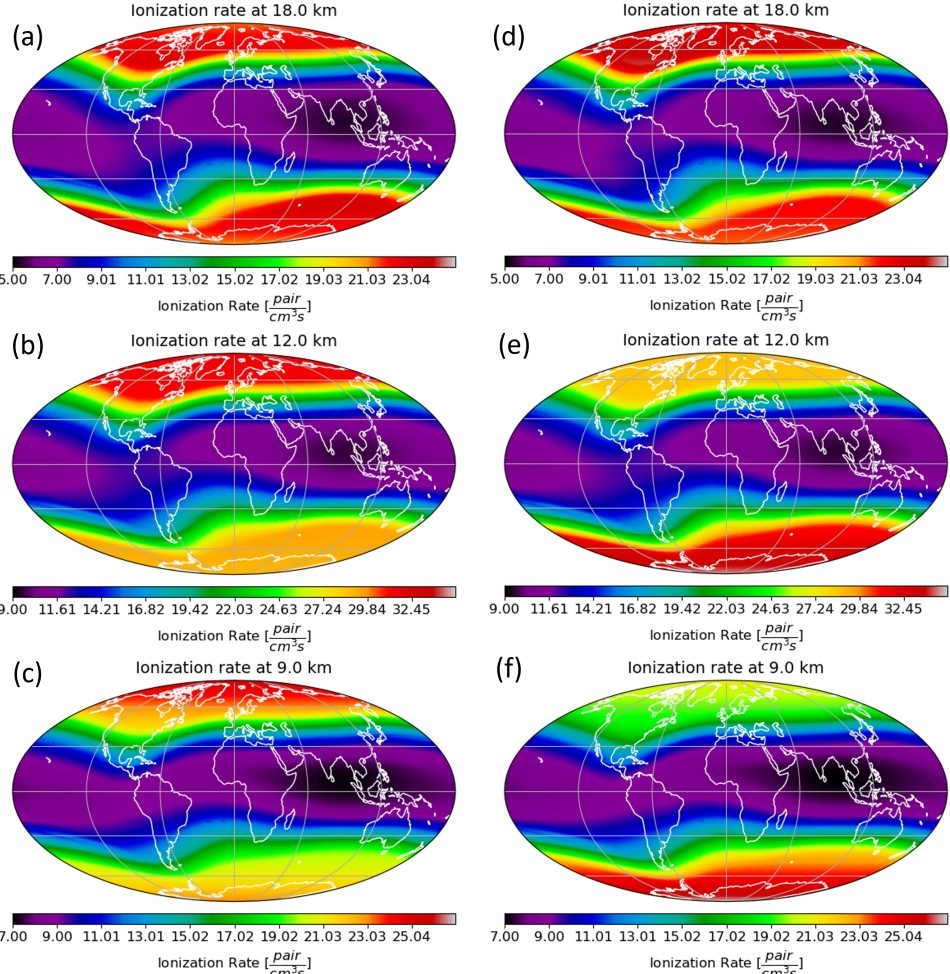

**Figure 7.** Global maps of the ionization rate (in $\frac{\text{pairs}}{cm^3s}$) computed in January (left panels) and August (right panels) at fixed altitudes, 18 km (**a,d**), 12 km (**b,e**), and 9 km (**c,f**). At a given altitude, the scale of the ionization rate is the same for January and August.

The situation is very similar for the absorbed dose displayed in Figure 8. As for the ionization rate, the absorbed dose is at its maximum in the polar regions during local winter, i.e., in the northern hemisphere in January and in the southern hemisphere in August. However, because the radiation dose does not go through a maximum at lower altitudes as the ionization dose does, and because it steadily decreases from high northern to high southern latitudes in January (the situation is reversed in August), the radiation dose is always found to be at maximum at the highest latitudes.

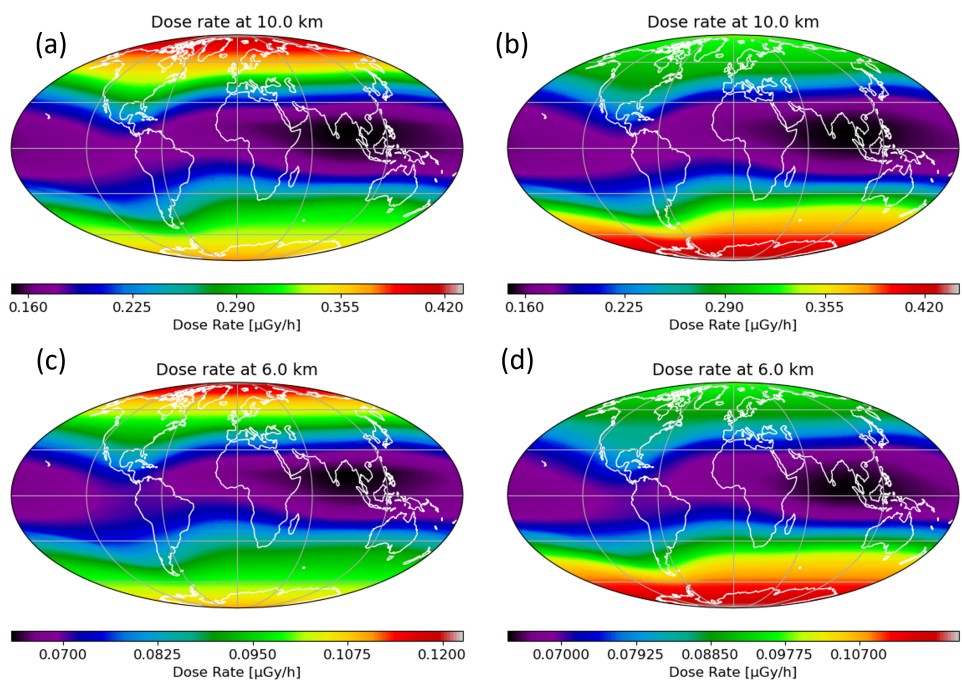

**Figure 8.** Global maps of the absorbed dose rate (in $\frac{\mu Gy}{h}$) by a water ICRU sphere computed in January (left panels) and August (right panels) at fixed altitudes, 10 km (**a**,**b**) and 6 km (**c**,**d**). At a given altitude, the scale of the ionization rate is the same for January and August.

## 4. Conclusions

In this work, we calculated the CR-induced ionization and absorbed dose rates within the Earth's atmosphere using AtRIS. We computed these quantities by utilizing different input atmospheric profiles (density, temperature, pressure, and composition) that were selected at various latitudes and for each month in 2014. As a first step, we investigated the latitudinal effect on the corresponding modeled profiles. To verify the effect of the atmosphere on atmospheric ionization and absorbed dose, the modulation potential and the cutoff rigidity values were fixed to 700 MV and 0 GV, respectively. The results of the simulations showed that ionization and absorbed dose rate profiles exhibit temporal and latitudinal variations. The relative difference between the profiles computed at the equator and different latitudes for January 2014 and the other months of 2014 shows a significant dependence on altitude. The largest variations for atmospheric ionization were found in the upper atmosphere, while the relative difference between the dose rate profiles remained negligible. At altitudes below 20 km, important differences in both the ionization and absorbed dose rates were found, which, in the case of the ionization dose, are less important than in the upper atmosphere. We further computed the global distributions of the ionization and dose rates induced by GCR protons and alphas, applying three simulation setups (i.e., changing latitude of the atmosphere that serves as an input for AtRIS). The modeled differences are significant. One of the highlights is the unexpected occurrence of an ionization maximum down to 50° N in August 2014, resulting from an interplay between the changes in altitudes of the Pfotzer maximum and the variation in the cutoff rigidity with latitude. Our findings, thus, highlight the necessity of dedicated modeling efforts, including the altitude, latitude, and season-dependent atmospheric density changes. Neglecting these changes, may result in an underestimation/overestimation of the CR-induced ionization of up to 68%, and in an underestimation/overestimation of the absorbed dose of about 60% at low altitudes. Moreover, the results of the simulations accounting for the seasonal variations in atmospheric density suggest that GCR-induced ionization and absorbed dose rate vary by up to 20% and 25%, respectively, over the course of one year at high latitudes in the lower part of the atmosphere.

The impact of GCRs on the atmosphere forms the ionization and absorbed dose baseline. Thus, it is reasonable to argue that above 20 km, where the relative difference between ionization profiles can reach up to 200%, is a region of the atmosphere where the ionization induced by GCRs is less significant, and thus may be considered irrelevant. However, accounting for ionization and absorbed dose rates induced by solar energetic particles (SEPs), which mostly affect high altitudes, the variations caused by the atmospheric density variations could become of significant importance. Because these variations that we show in this work are caused by the state of the atmosphere and not by a change in the primary particle spectrum, we expect such variations to be maintained in the case of an SEP event, a study that will be performed in the future.

**Author Contributions:** Conceptualization, A.W., V.P. and E.B.; methodology, A.W.; software, A.W.; validation, A.W., V.P. and E.B.; formal analysis, A.W.; investigation, A.W.; writing—original draft preparation, A.W.; writing—review and editing, A.W., V.P., E.B. and K.H.; visualization, A.W.; supervision, V.P. and E.B.; project administration, V.P.; funding acquisition, V.P. All authors have read and agreed to the published version of the manuscript

**Funding:** This research was funded by the European Partnership on Metrology (grant no. 21GRD02 BIOSPHERE).

**Institutional Review Board Statement:** Not applicable.

**Informed Consent Statement:** Not applicable.

**Data Availability Statement:** The atmospheric response matrices created during this study are available from the corresponding author on reasonable request.

**Acknowledgments:** The project 21GRD02 BIOSPHERE has received funding from the European Partnership on Metrology, cofinanced by the European Union's Horizon Europe Research and Innovation Programme and by the participating states. KH acknowledges the European Union's Horizon 2020 research and innovation programme under grant agreement No. 870405 and thanks Bernd Heber and Saša Banjac (Christian-Albrechts-Universität zu Kiel) for maintaining AtRIS.

**Conflicts of Interest:** The authors declare no conflict of interest.

## Abbreviations

The following abbreviations are used in this manuscript:

| | |
|---|---|
| AtRIS | Atmosphere Radiation Interaction Simulator |
| PSF | Planet Specification File |
| CRII | Cosmic-Ray-Induced Ionization |
| GCRs | Galactic Cosmic Rays |
| SEPs | Solar Energetic Particles |

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
