# Peer review of "The Atmospheric Influence on Cosmic-Ray-Induced Ionization and Absorbed Dose Rates"

_universe, doi:10.3390/universe9120502_

Round 1

Reviewer 1 Report

Comments and Suggestions for Authors

Reviewer report on the manuscript “The atmospheric influence on cosmic ray induced ionization and absorbed dose rates ” by Alexandre Winant  et al.  The manuscript discusses variability of ionization and dose rates induced by GCR into atmosphere for various seasons and latitudes. The manuscript may be interested for scientific community if correct computations and analysis of the obtained results will be done. Now manuscript needs in results with more simulations and visualization of the obtained results. The structure of manuscript should be changed. The discussion part of the manuscript should be included into the parts of the manuscript where ionization and dose rates induced by GCR are discussed. All effects into atmosphere from ionization and dose rates induced by GCR as well as for other energetic particles mostly dependent on atmospheric densities. Computation of ionization and dose rates without atmospheric densities will be dependent only on yield functions of monoenergetic fluxed of particles, on energy particle depositions.

Main issues:

1) One of the main points for this study must be discussion of atmospheric densities before discussion of the computation results of ionization and dose rates.

2) The latitudinal variability of ionization and dose rates induced by GCR strongly dependent on cutoff rigidity as well as effects at latitudes depends on cutoff rigidity various from 0 GV over polar region to about 18 GV over equator.  In present study the authors just used one cutoff rigidity Rc=0 GV and try do discuss ionization and dose rates induced by GCR over various latitudes. The computation results should be repeated with correct cutoff rigidity various from 0 GV over polar region to about 18 GV over equator.

3) The authors should discuss energy range that considered as particles of cosmic rays that includes energy range of GCR spectra recovered by neutron monitors and energy particles obtained by satellites as alpha particles, magnetospheric protons or particles during SEP. As I could see in this study the authors considered energy particles flux that seems to be covered energies of precipitated particles detected by satellites and neutron monitors. This discussion is very important for explanation of atmospheric effects because ionization and dose rates usually recovered only based on spectra of neutron monitors and ionization effects really can be obtained till equatorial regions. The ionization rates in the atmosphere based on neutron monitor GCR spectra is at altitudes from about ground till about 60 km. In this manuscript all energies are mixed and reader can think that all particles will precipitate into the whole atmosphere over all latitudes and lead to ionization rates over equatorial region.

It is not correct from physical point of view lead a discussion of the computation results of ionization and dose rates induced by GCR over altitudes higher than about 50 km at latitudes over equatorial region because of photons and alphas with energies less than about 500 MeV cannot precipitate in this region. I suggest authors check real energies of precipitated particles into atmosphere observed for example by POES or GOES satellites and real spectra of GCR obtained by neutron monitors with various cutoff rigidities. The results of author’s computation into atmosphere must be compared with results discussed in the literature at least and should based on correct spectra estimation.

Issues:

Line 41: “atmospheric ionization down 20 km” Add references, for example, Ionization effect of solar particle GLE events in low and middle atmosphere by Usoskin et al, 2011.

Line 64: Why for investigation was chosen 2014 year? Maximum in solar cycles were also in 2001, 1989 and in other years.

Figure 1:  Why do you need this figure, what it can be explained and how it is used. Figure 1 should be deleted. Moreover in not clear why atmosphere should be divided on boxes [90°N, 30°N], [30°N, 30°S] and [30°S, 90°S] if at the end considered ionization and dose rates just on specified latitudes 85, 45, 15, 0 in northern and southern hemispheres. Add explanation into the text of the manuscript.

Line 92: What mean “of 200 layers” how altitudinal steps can describe it? Add explanation into the text of the manuscript.

 Line 127, Fig.1: The figure does not informative; instead of this figure add atmospheric densities profiles as input profiles for Figure 2.

Figure2: This figure does not correctly represent the deposition of energetic particles over the equatorial region of atmosphere. Protons and alpha particles with energies less then about 500 MeV cannot precipitate in equatorial region because cutoff energy rigidity. The authors need to present the correct precipitation of energetic particles for 2014 based on real spectra obtained from satellites that detects precipitated particles into atmosphere and from neutron monitors that measure particles of GCR that already pass through atmosphere, and then it will become clear that the modeled matrix is incorrect from a physical point of view. Correct Figure 2 in the manuscript.

Line 130: What mean atmospheric profiles taken at ±15°, ±45° and ±80° ? It is not clear from Figure1. I recommend delete Figure 1 and add really atmospheric densities that correspond investigated latitudes.

Lines 144-147: Where it can be seen? If you use different modulation potentials it is not clear why do need computation for 2014 year?

Lines 155-157: Where it can be seen? In the present manuscript the results presented only until 80°N to 80°S.

Line 169 and Figure 3 : Why Rc=0 GV but not others cutoff rigidities if altitudinal effects are considers? Why 700 MV correspond to 2014 year, the authors should add references.

Figure 3: Profiles at various latitudes should be characterized by various cutoff rigidity, but you consider only one Rc=0 GV. How you computed effects on various latitudes?

Figure 3: Atmospheric density profiles for the corresponding latitude should be added here. How do you explain from physical point of view so strong difference in ionization rates between two hemispheres without atmospheric densities profiles? Here authors just described what they see but they must add text why it happened based on atmospheric densities profiles. Here authors must compare their results with computation ionization rates based on spectra of precipitated particles like as protons and alphas particles detected by satellites for example POES (energies below 300 -500 MeV) and ionization GCR (energies higher 300 -500 MeV) based on spectra from neutron monitor.  The authors must exclude Figure 9 from section discussion and add this Figure as additional panels to Figure 3 and to Figure 6.  The right panel of Figure 3 does not clear from scientific point of view. Why needs visualization of deviation of ionization rates from equator? The right panel of Figure 3 must be deleted because it is not informative and does not lead to any scientific importance.

Lines 169 and 219: Figure 4 and Figure 5 – the same problem as with Figure 3.  Profiles at various latitudes should be characterized by various cutoff rigidity, but you consider only one Rc=0 GV. How you computed effects on various latitudes?

Figure 5: the left and right panels should have the same altitudes.

Lines 221 etc.: The authors just describe figures and do not discuss the reason of these differences. Add figures with atmospheric densities during investigated time/altitudes and seasons. The authors should discuss atmospheric densities over various altitudes and only after that can describe ionization rates.

Figure 6: the same problem as with other figures. Profiles at various latitudes should be characterized by various cutoff rigidity, but you consider only one Rc=0 GV. How you computed effects on various latitudes? Moreover here computation was done for January 2014, as it is seen in the title of figure, but results in figure present ionization and dose rates over various months. How it can be?

The authors must exclude Figure 9 from section discussion and add part of Figure 9 as additional panels to Figure 6.

Lines 243-270 etc: The same situation as with previous figures. The authors just describe figures and do not discuss the reason of these differences. Add figures with atmospheric densities during investigated time/altitudes/latitudes/seasons and properly discuss the variability of ionization rates and dose rates. The authors do not need to discuss ionization and dose rates without discussion atmospheric densities. The differences are connected with properties of atmosphere.

Lines 284- 310: Figure 7 and Figure 8– the same problem as with other figures. Add atmospheric densities at discussed altitudes in the same way as ionization and dose rates (fig.7 and fig 8) during January and August. You should discuss atmospheric densities over various altitudes/time and only after that you can describe ionization and dose rates.

Line 312: Exclude Figure 9 from section discussion and add this Figure as additional panels to Figure 3 and to Figure 6.

Lines 323-328. Add explanation of this difference from physical point of view.

Lines 333-337: Move this explanation to the figure where it is shown. Now the explanations are not clear without figures.

Lines 338-342: Move this explanation to the figure where it is shown. Now the explanations are not clear without figures.

Line 344-345: Where it is seen the both polar hemispheres are similar? Take real data and confirm this sentence based on atmospheric densities over both polar hemispheres. Add figures with densities and move this explanation to the figure where ionization and dose rates discussed.

Line 344-361: Move this explanation to figures where it is shown. Now the explanations are not clear without figures. Add figures with discussed atmospheric densities.

Line 397-398: Where long-term simulations of ionization and dose rates are shown? Before doing conclusion the authors should discuss results in the manuscript, show atmospheric densities for this period of time and only after that discuss ionization and dose rates. The authors can add results or must exclude this conclusion.

Line 404-409. The same situation presents as with conclusion in lines 397-398. Where SEP simulation results are? What about energy range for SEP? Before doing conclusion the authors should discuss results in the manuscript, show SEP spectra, atmospheric densities for period of SEP and only after that discuss ionization and dose rates. The authors can add results or must exclude this conclusion.

From physical point of view the statement include into lines 404 – 406 are not correct as soon as authors do not say what means upper atmosphere, what is class of SEP, what is spectra, energy range of particles into the SEP flux and intensity of SEP flux.  Please read more careful results presented by papers of Usoskin I., Mishev A., Jackman Ch., Wissng J.M. etc.  before doing your own investigation.

Reviewer 2 Report

Comments and Suggestions for Authors

Overall the article is reasonably well written with clear objectives, methodology and results well described and well defined, respectively; and in line with the objectives.
I consider this work to be of great relevance due to the impact of its results on the description of radiation absorbed by the atmosphere, especially the impact on air transport.

Therefore, there are some questions that need to be better clarified:

a) What is the basis for this atmospheric composition? (78.084% N2, 20.947% O2, 0.934%Ar, 0.000524% He at the surface.) It is not clear that this composition will be used or changed at different altitudes.
between 0km and 100km the composition is substantially different. lines 92-93.

b) Were the particles used only protons and alpha? This needs to be made clearer. Mainly in relation to the energy of the primary particle, since for energies above 10^17 eV the mass spectrum is not well defined.

c) What is the energy range of the primary particles considered in the simulation? It is not clear from the text.

d) In the profiles shown in figures 3, 4 and 6 it is not clear where the errors associated with each profile are. Were systematic simulations carried out? This is also not clear from the text.

e) What would be the impact of these errors on seasonal variations, shown in figure 5, and on global maps, shown in figures 7 and 8?

Minor fixes:

a) despite being mentioned in the text, the description of the three sub-atmospheres of [90°N, 30°N], [30°N, 30° S] and [30°S, 0°S] is not explicit in figure 1 I suggest explaining the division in the same figure or adding a new figure.

Comments on the Quality of English Language

No comments !

Round 2

Reviewer 1 Report

Comments and Suggestions for Authors

Reviewer's report on the manuscript "The atmospheric influence on cosmic ray induced ionization and absorbed dose rates" by Alexandre Winant et al. 

The authors answered most of the reviewer's questions, but some questions still remain open. After taking into account the comments, the manuscript can be accepted for publication.

1) None of the publications are visible in the latest version of the manuscript. It is necessary to properly compile the technical file taking into account the bibliography.

2) Line 116 "One phantom provided in AtRIS is an ICRU sphere (defined  by the International Commission on Radiation Units)" - add here reference or link to simulation tool.

and

Line  386 "a water ICRU 386 sphere without shielding" - add here reference or link to simulation tool.

3) Add references and a few sentences to your manuscript about how the radiation dose is calculated to obtain the results.

Reviewer 2 Report

Comments and Suggestions for Authors

The questions were all clarified and implemented in the article. For my part, I consider everything corrected and ready for publication.

Author Response

We thank you for your helpfull comments and suggestions.